# New Insights on Gene by Environmental Effects of Drugs of Abuse in Animal Models Using GeneNetwork

**DOI:** 10.3390/genes13040614

**Published:** 2022-03-29

**Authors:** Alisha Chunduri, Pamela M. Watson, David G. Ashbrook

**Affiliations:** 1Department of Biotechnology, Chaitanya Bharathi Institute of Technology, Hyderabad 500075, India; msd.alisha07@gmail.com; 2Department of Genetics, Genomics, and Informatics, University of Tennessee Health Science Center, Memphis, TN 38163, USA; pwatso16@uthsc.edu

**Keywords:** addiction, human genetics, systems genetics, gene expression, bioinformatics, cross-species analysis, animal models, genomics, genetic mapping

## Abstract

Gene-by-environment interactions are important for all facets of biology, especially behaviour. Families of isogenic strains of mice, such as the BXD strains, are excellently placed to study these interactions, as the same genome can be tested in multiple environments. BXD strains are recombinant inbred mouse strains derived from crossing two inbred strains—C57BL/6J and DBA/2J mice. Many reproducible genometypes can be leveraged, and old data can be reanalysed with new tools to produce novel insights. We obtained drug and behavioural phenotypes from Philip et al. Genes, Brain and Behaviour 2010, and reanalysed their data with new genotypes from sequencing, as well as new models (Genome-wide Efficient Mixed Model Association (GEMMA) and R/qtl2). We discovered QTLs on chromosomes 3, 5, 9, 11, and 14, not found in the original study. We reduced the candidate genes based on their ability to alter gene expression or protein function. Candidate genes included *Slitrk6* and *Cdk14*. *Slitrk6*, in a Chromosome14 QTL for locomotion, was found to be part of a co-expression network involved in voluntary movement and associated with neuropsychiatric phenotypes. *Cdk14*, one of only three genes in a Chromosome5 QTL, is associated with handling induced convulsions after ethanol treatment, that is regulated by the anticonvulsant drug valproic acid. By using families of isogenic strains, we can reanalyse data to discover novel candidate genes involved in response to drugs of abuse.

## 1. Introduction

Two of the biggest problems in analyses of biomedical data are irretrievability and irreplicability. Biomedical data is often lost as soon as it is published, locked within a forgotten hard drive, or siloed in a little-used format on a lab’s website. There are many efforts to make data publicly accessible and retrievable, such as the FAIR Principles (findability, accessibility, interoperability, and reusability) [1], and these allow the combined analysis of many datasets and reanalysis using new tools. There is still the problem of irreproducible datasets: for example, if a sample from a particular outbred cohort is found to be an outlier during data analysis, there is no way to go back to that genometype and remeasure the phenotype. Nor can new phenotypes be measured in the same individuals within the same environments later as new tools emerge. The genometype refers to all genotype states across the organism. Different strains in the BXD family might share the same genotype at a specific location, but the different strains are different genometypes. Families of isogenic strains solve this problem, allowing for reproducible genometypes that can be sampled many times, under many environmental conditions, leading to so-called experimental precision medicine [2]. This means that a genometype sampled 30 years ago in a different country can be replicated now, in any lab, with any environmental variable of interest, using any technique. The GeneNetwork.org (http://www.genenetwork.org/, accessed on 2 February 2022) website allows this combination of FAIR data and reproducible genomes, meaning that research teams can now go back to previous datasets and reanalyse them with new data and new tools. Every new dataset adds exponentially to the number of possible connections. In this paper, we will reanalyse drug and addiction related data from over a decade ago, using new genometypes for the BXD family of murine strains, as well as new statistical tools, showing that we can identify new quantitative trait loci (QTLs), resulting in highly plausible candidate genes.

Quantitative trait locus (QTL) mapping has been carried out in numerous species to associate regions of the genome to phenotypes even before the structure of the genome was well understood (e.g., [3]). Rodents, especially mice, have been the species most prominently used for biomedically relevant traits. Amongst these, the BXD family of recombinant inbred (RI) strains derived from crossing two inbred strains—C57BL/6J and DBA/2J mice—have been extensively used for almost 50 years in fields such as neuropharmacology [4,5,6], immunology [7,8,9,10,11,12,13], behaviour [13,14,15,16,17,18,19,20,21], aging [21,22,23,24,25,26,27,28,29], neurodegeneration [30,31,32,33], and gut microbiome–host interactions [34].

The development of the BXD panel was started by Benjamin A. Taylor by inbreeding the progeny of female C57BL/6J and male DBA/2J strains, for the purpose of mapping Mendelian traits [35]. This led to the original 32 BXD strains, which now carry the suffix ‘TyJ’ (Taylor to Jackson Laboratory). To increase the power and precision of QTL mapping, the number of strains has been expanded [36], including through advanced intercross [37], to a total of 140 extant strains [2], making this resource the largest family of murine isogenic strains. Phenotypes in the BXD have been measured under many conditions, allowing for the identification of gene-by-environment interactions. Understanding these interactions can potentially help in the discovery of complex therapeutic solutions and are a vital part of the development of precision medicine.

GeneNetwork.org is a tool for quantitative genetics that started in 2001 as WebQTL [38]. It evolved from analyses of forward genetics in the BXD mouse family, to phenome-wide association studies and reverse genetics in a variety of species. Although GeneNetwork.org contains data for many species and populations, it most prominently contains data for the BXD family. Over 10,000 “classical” phenotypes, measured under a variety of environmental conditions and over 100 ‘omics datasets, are available on GeneNetwork.org for the BXD family. GeneNetwork.org and the BXD RI population are therefore a powerful tool for systems genetics and experimental precision medicine. The great advantage of inbred lines, with stable genometypes that can be resampled is that data can be reused and reanalysed over time, as tools improve. From the very start of the genome sequencing revolution, when loci were first mapped to causative genes, new tools and a greater understanding of the genome have allowed us to go back to old data and gain new insight.

In this study, we will demonstrate how new biological insight into drugs of abuse can be gained by reanalysing data in the BXD family, using improved genometypes from sequencing, and new mapping methods (linear mixed models). Using this method, we have discovered new QTLs and candidate genes for behavioural phenotypes associated with the predisposition of drug- and behaviour-related traits obtained from Philip et al. 2010 [39].

## 2. Materials and Methods

### 2.1. Phenotype Data

The traits used for analysis in this study were acquired by Philip and team and published in 2010 [39]. All datasets from this publication are freely available on GeneNetwork.org, and were obtained from the BXD published phenotypes (http://gn1.genenetwork.org/webqtl/main.py?FormID=sharinginfo&GN_AccessionId=602&InfoPageName=BXDPublish, accessed on 2 February 2022). The original study aimed to determine the influence of genes in response to the environment and the plausibility of similar interactions with drug-related attributes including response to and withdrawal from cocaine, 3,4-methylenedioxymethamphetamine, morphine, and ethanol and the correlation to phenotypic traits including anxiety, locomotion, stress sensitivity, and pain sensitivity. Complex phenotyping batteries consisting of diverse behavioural assays were employed on the RI strains, and multivariate analyses were performed using GeneNetwork.org. An interplay between environmental factors, drug-induced neural changes, and genetic factors underlies the predisposition of an individual to addiction. In this study, a total of 762 traits were analysed (Appendix A) using new genotypes and linear mixed model (LMM) based mapping software, to identify novel candidate genes and gene-by-treatment interactions. However, we did not include morphine related traits, as these are being actively studied by others. Of the then extant population of 79 strains [7], Philip’s study used approximately 70 strains to measure the traits. 

### 2.2. New Genotypes from Sequencing

A total of 152 BXD strains have now been sequenced using linked-read technologies, and new genotypes for all 152 BXD strains have been produced from this (European Nucleotide Archive project PRJEB45429). Variants were chosen to define the start and end of each haplotype block, and variant positions from the previously published genotypes were kept allowing maximum back compatibility with previous publications.

### 2.3. Genome-Wide Efficient Mixed Model Association (GEMMA), Kinship within the BXD Strains, and QTL Mapping

The BXD family has been produced in several ‘epochs’ across 40 years, using both standard F2 recombinant inbred methods as well as advanced intercross recombinant inbred methods [2]. This has led to both expected and unexpected kinship between BXD strains. This kinship between strains can lead to bias, as it breaks the expectations of previously used methods, such as the Haley–Knott mapping algorithm that was used in the original study. Updated linear mixed models including R/qtl2 (qtl2 analysis using R software) and Genome-wide Efficient Mixed Model Association (GEMMA), which is accessible in GeneNetwork.org, have been used for this study as they allow correction for kinship, as well as other cofactors if needed.

An analysis of 762 traits taken from Philip et al. study was carried out using the GEMMA mapping tool with the genotypes from sequencing, a minor allele frequency (MAF) of 0.05, and utilizing the Leave One Chromosome Out (LOCO) method. This computation provides a −log(p) value between each marker and the phenotype. We used a −log(p) > 4, as significant. However, since permutations of the GEMMA algorithm are not currently available in GeneNetwork.org, we confirmed the significance of these QTL using the linear mixed model tool within R/qtl2 [40], with 5000 permutations of the data. 

### 2.4. Identification of Novel QTLs

Two methods were used to identify significant QTLs. Firstly, traits with an adjusted *p* < 0.05 using permutation in R/qtl2 (described above) were investigated at length, as these are significant after empirical correction. The second method used was to take advantage of independent traits which share QTL at the same location with suggestive *p*-values (*p* < 0.63). This *p* < 0.63 equates to one false positive per genome scan. However, the likelihood of any chromosome having a QTL on it is approximately 1 in 20 (i.e., *p* < 0.05) due to 20 chromosomes in mice. The likelihood of two independent traits sharing the same QTL location by chance is therefore much lower than *p* < 0.05. Traits were referred to as independent if they were carried out in separate groups of animals (e.g., males and females), or if the traits were measured at independent timepoints (e.g., at 10 min after treatment and 60 min after treatment).

### 2.5. QTL Confidence Intervals

A 1.5 LOD (logarithm of the odds) or 1.5 −log(p) drop [41] was used to determine the QTL confidence interval for each statistically significant trait (in our case of a two-parent population LOD and −log(p) are approximately equal). Therefore, for each of the QTL above (Appendix A), we were able to generate a list of genes within this confidence interval. Genes were called within the QTL interval using the GeneNetwork.org QTL mapping tool, which provides protein coding genes, non-coding genes, and predicted gene models. 

### 2.6. Cis-eQTL Mapping

A cis-eQTL indicates that a variant within or very close to a gene influences its expression. Genes with cis-eQTLs are high priority candidates, as it provides a potential causal pathway between the gene variant and the phenotype of interest (i.e., the variant alters gene expression, and the expression of that gene alters the phenotype). Therefore, if a gene within a QTL interval is cis-regulated, we categorize it as a high priority candidate. For each QTL, we identified which, if any, genes within the QTL interval also had a cis-eQTL, and in which tissues an eQTL was seen (using transcriptome data from GeneNetwork.org). Using this same data, we also identified correlations between expression of each of these genes and the phenotype of interest.

### 2.7. “Gene Friends”, or Co-Expression Analysis

Genes with a *cis*-eQTL in at least one tissue were further considered for co-expression analysis. The top 10,000 correlations were generated in the tissue with the highest correlation between gene expression and the phenotype of interest. Gene-gene correlations with Sample *p*(r) < 0.05 were taken into WebGestalt to perform an over-representation analysis [42,43,44,45]. This results in the identification of significantly enriched annotations or pathways in the genes which co-express with our gene of interest. This can often suggest pathways or networks that the gene is involved in, even if the gene itself has not yet been annotated as part of that network.

### 2.8. Gene Variant Analysis

Deep, linked-read sequencing of the 152 members of the BXD family was carried out using Chromium 10X sequencing (https://www.10xgenomics.com/products/linked-reads, accessed on 2 February 2022), resulting in 5,390,695 SNPs and 733,236 indels, which are high confidence and segregate in the population (i.e., have a minor allele frequency greater than 0.2). These 6 million variants are potential causes of QTLs detected in the BXD family.

To identify potential effects of these variants, we used the Variant Effect Predictor (VEP) website (http://ensembl.org/Tools/VEP, accessed on 2 February 2022 [46]). All variants within our QTL intervals were extracted from the variant VCF file and uploaded to the VEP. Potentially deleterious variants or variants which impact protein function were identified using the “Consequence”, “IMPACT”, “SIFT” [47,48] and “BLOSUM62” [49] annotations.

### 2.9. PheWAS

Phenome-wide association studies (PheWAS) utilize a genomic region of interest to find associations between that region and phenotypes measured in GWAS datasets. We used human PheWAS data for all the candidate genes in our QTLs to detect genes with relevant human phenotype associations (i.e., behavioural and neurological phenotypes). A relevant association implies confidence in a candidate gene and suggests cross-species translatability of the finding. We used online PheWAS tools, GWASatlas (https://atlas.ctglab.nl/PheWAS, accessed on 2 February 2022, [50]) and PheWeb (http://pheweb.sph.umich.edu/, accessed on 2 February 2022) for this study.

## 3. Results

### 3.1. Identification of QTLs

We first sought to identify novel genetic loci linked to the phenotypes from Philip et al., 2010 [39] that were not found in the original study. Comparing QTL mapping using Haley–Knot (H-K; as used previously) [51] and GEMMA, there are 426 traits which had a maximum LRS < 17 with H-K (i.e., non-significant), that now have a maximum −log(p) > 4. These new QTL are therefore of interest (Table 1). To confirm these, we performed linear mixed model (LMM) QTL mapping in R/qtl2, with permutations. This produced 61 traits which are significant compared to the empirical significance threshold generated by permutations (Appendix A).

Two methods were used to identify QTLs of interest. First, the group of 61 traits that were significant by permutations were analysed. The second method was to take advantage of independent traits which share QTL at the same location with suggestive *p*-values (*p* < 0.63). Traits were referred to as independent if they were carried out in separate groups of animals (e.g., males and females) or if the traits were measured at independent timepoints (e.g., at 10 min after treatment and 60 min after treatment). We identified 25 QTL for 267 traits (Appendix A).

### 3.2. Novel QTL

For each of the QTL identified above, we determined if they were reported in Philip et al.’s original study [39], or if related phenotypes have been reported in the MGI database [52].

Several locomotion traits related the QTL map to Chr1:37.671–78.94 Mb (Figure 1) that were not detected in the Philip et al. study. Previously detected relevant phenotypes associated with this region include the loss of righting reflex induced by ethanol [53] and vertical clinging [54].

We report a novel QTL on Chromosome3 (51.723–56.473 Mb) for vertical clinging activity, and on Chromosome4 (105.245–114.11 Mb) for locomotion in response to cocaine. Previous studies show a QTL for anxiety in this region of Chromosome4 [55]. We also report novel QTLs on Chromosome5 for handling induced convulsions as an ethanol response (4.468–5.172 Mb) as well as locomotion in response to cocaine (99.801–101.331 Mb). Finally, there was a novel QTL for locomotion in response to cocaine on Chromosome11 (46.361–50.383 Mb) (Appendix A).

### 3.3. Candidate Causal Genes within Novel QTL

We concentrated on a subset of six novel QTL that contained less than 100 genes. These QTLs are more amenable to finding plausible candidate genes using bioinformatic methods. After reducing the likelihood of finding false positives, these large QTLs are more likely to be due to two or more variants in different genes both contributing to the phenotype. The advantage of families of isogenic strains of mice, such as the BXD, is that more strains could be phenotyped, reducing the size of these QTL regions and allowing for greater precision. We leave these large QTLs to future studies. The smaller QTL regions investigated here were: Chr3:51.723–56.473 Mb, Chromosome5:4.468–5.172 Mb, Chr5:99.801–101.331 Mb, Chr9:45.671–48.081 Mb, Chr11:62.923–65.082 Mb and Chromosome14:109.994–114.751 Mb (Appendix A)

We used several tools to narrow down potential candidate genes within these QTLs. Variants can change phenotype in two main ways: they can either change gene expression or can change protein function. To look for variants altering gene expression, we first looked for genes within our QTL regions with local or cis-eQTL. Cis-eQTL demonstrate that there are variants in or close to a gene that cause changes in that gene’s expression. This is useful, since it clearly shows that a variant in the eQTL region has a regulatory effect. Therefore, genes with a cis-eQTL are interesting candidate genes.

The next step is to investigate whether the expression of these genes correlates with the phenotype(s) of interest. This would suggest a chain of causality: a variant within a gene causes a change in its expression, and the expression of that gene correlates with expression of a phenotypic trait of interest. To do this, we created a correlation matrix between all genes within a QTL with a cis-eQTL in any brain tissue as well as the phenotypes that contributed to the QTL (Appendix A). Any gene with a cis-eQTL and a significantly correlated expression was considered a good candidate. If the gene only had a cis-eQTL and correlation in a single brain region, then it suggested that this brain region might also be of interest for the phenotype (adding another link to this chain).

The QTL region for vertical activity (Chromosome3 51.723–56.473 Mb) has 60 genes among which six genes have cis-eQTLs (Figure 2) (Appendix A). No relevant functional annotations (Gene Ontology) have been reported. Dclk1 (location of cis-eQTL: Chromosome3 55.52 Mb) variants were previously reported to be associated across Schizophrenia and Attention Deficit Hyperactivity Disorder [56]. The same gene has been described as a candidate gene for inflammatory nociception [57]. *Trpc4* (location of cis-eQTL: Chromosome3 54.266176 Mb) may be involved in the regulation of anxiety-related behaviours [58].

The QTL region for handling induced convulsions (ethanol response; Chromosome5 4.468–5.172 Mb) house two genes (*Fzd1* and *Cdk14*) with cis-eQTLs among the three present in this region. *Fzd1* (location of cis-eQTL: Chromosome5 4.753 Mb) receptor regulates adult hippocampal neurogenesis [59]. The QTL corresponding with locomotion in response to cocaine (Chromosome5 99.801–101.331 Mb) has ten genes with cis-eQTLs (Figure 3). QTL analysis of *Enoph1* (location of cis-eQTL: Chromosome5 100.062 Mb) in mice indicates that it plays a role in stress reactivity [60]. Variants of *Coq2* (location of cis-eQTL: Chromosome5 100.654 Mb) contribute to neurodegenerative disorders such as Parkinson’s disease [61].

Relevant annotations for other genes with cis-eQTLs have not been reported yet by other studies. The QTL corresponding with mechanical nociception (Chromosome9 45.671–48.081 Mb) includes five genes with cis-eQTLs. *Sik3* (location of cis-eQTL: Chromosome9 46.222 Mb) is involved in regulating NREM sleep behaviour in mice [62]. *Cadm1* (location of cis-eQTL: Chromosome9 47.550 Mb) knockout mice show increased anxiety, impaired social and emotional behaviours, and disrupted motor coordination (Figure 4) [63].

An analysis of the locomotion in response to cocaine QTL (Chromosome11 62.923–65.082 Mb) revealed five genes with cis-eQTLs. *Arhgap44* (location of cis-eQTL: Chromosome11 65.005456 Mb) has phenotype associations related to abnormal motor learning, abnormal response to novel objects, increased grooming behaviour, and hypoactivity (Figure 5) [64]. The brain regions with highest correlation have been added to the Supplementary Information (Appendix A). The QTL corresponding to locomotion in the centre (Chromosome14 109.994–114.751 Mb) has a single gene with a cis-eQTL, *Slitrk6* (location of cis-eQTL: Chromosome14 109.231826 Mb, in Figure 6). Knockout of this gene has been associated with impaired locomotory behaviour and altered responses to a novel environment, making this gene a strong candidate [65].

### 3.4. Co-Expression Networks or “Gene-Friends”

Genes that are co-expressed are often part of the same pathways or networks which contribute to similar phenotypes. These so-called ‘gene-friends’ [66,67] can provide insights into the function of an unannotated gene as function can be implied from the known functions of the genes it co-expresses with. As new datasets are being generated for the BXD consistently (now including methylation, proteomic and metabolic datasets), new associations can be found.

For each of the genes within our phenotype QTLs that also has a cis-eQTL in at least one dataset on GeneNetwork.org, we performed a correlation analysis with all other probes or genes within that dataset. We then performed an enrichment analysis using WebGestalt using all the probes or genes that correlated with our gene of interest (i.e., the gene with a cis-eQTL), and investigated if any of the enriched annotations or pathways were relevant to the phenotype.

Highly relevant enriched phenotypes were found in genes co-expressing with *9430012M22Rik* (location of cis-eQTL: Chromosome3 55.291 Mb). This gene is present in a QTL for vertical activity (BXD_12023). Genes that correlate with expression of *9430012M22Rik* in the neocortex are enriched for involvement in abnormal locomotor behaviour (FDR = 1.2056 × 10^−9^) and abnormal voluntary movement (FDR = 7.1848 × 10^−10^). Other results for this co-expression network that may be relevant include abnormal synaptic transmission and abnormal nervous system physiology (Appendix A). The genes that correlate with expression of *BC033915* (location of cis-eQTL: Chr9 45.671–48.081 Mb) in the hippocampus are enriched for abnormal motor capabilities/coordination/movement (FDR = 2.3483 × 10^−11^). Other relevant results include abnormal brain morphology and abnormal nervous system physiology. Similarly, genes in the *Slitrk6* co-expression network (location of cis-eQTL: Chromosome14 109.231 Mb) in the striatum are involved in abnormal locomotor behaviour (FDR = 6.978 × 10^−12^) and abnormal voluntary movement (FDR = 2.9352 × 10^−11^). This makes sense, since this Chromosome14 QTL is for locomotion.

Other genes with cis-eQTLs had significant enrichments that include abnormal brain morphology, abnormal body composition and abnormal nervous system physiology (Appendix A).

### 3.5. Gene Variant Analysis

The second method by which a variant can alter a phenotype is changing the protein structure or function. To examine this, we took advantage of the deep sequencing available for all BXD strains. We identified over 6 million common SNPs and small INDELs which segregate within the BXD family (i.e., occur in greater than >20% of the population). For each of the 6 QTL identified above, we looked for variants that were predicted to alter protein structure or splicing, or predicted to be deleterious by SIFT or BLOSUM, using the variant effect predictor (VEP).

The QTL located at Chr3:53.667–54.942Mb for vertical activity contains predicted deleterious variants in 10 genes (Table 2): two missense variants in *Ccdc169*; one missense variant in *Ccna1*; one in-frame insertion in *Dclk1*; two frameshift variants, a stop loss, and nine missense variants in *Frem2*; a frameshift variant and six missense variants in *Mab21l1*; a frameshift variant, a missense variant, eight frameshift variants, two in-frame deletions, 18 missense variants, and three stop losses in *Nbea*; four in-frame deletions, an in-frame insertion, six missense variants, and a start loss in *Postn*; a frameshift variant, eight missense variants, a stop gain, and a stop loss in *Spg20*; and a missense variant in *Trpc4*.

The Chr5:4.468–5.172Mb QTL for handling-induced convulsion in response to ethanol contains two missense variants in *Fzd1*, and four missense variants in *Cdk14*. The Chr5:100.164–100.895Mb QTL for cocaine related phenotypes, contains predicted deleterious variants in 8 genes (Table 3): three frameshift variants and three missense variants in *Cops4*; a missense variant and a stop-gain in *Enoph1*; a frameshift variant and five missense variants in *Hnrnpd*; three missense variants and a splice donor variant in *Hnrnpdl*; two frameshift variants, an in-frame insertion, 13 missense variants, and a splice donor variant in *Hpse*; three missense variants in *LIN54*; a frameshift variant, one in-frame deletions, and a splice donor variant in *Sec31a*; and a missense variant and a splice donor variant in *Tmem150c*.

The QTL located in chromosome 9 at 45.671–48.081Mb for mechanical nociception contains predicted deleterious variants in three genes (Table 4): A frameshift variant, two missense variants and a stop loss in *4931429L15Rik*; two frameshift variants and five missense variants in *Cadm1*; and an in-frame deletion, three missense variants, and a stop loss in *Cep164*. The Chr11:62.923–65.082Mb QTL for nociception contains four frameshift variants, an in-frame deletion, and thirteen missense variants in *Myocd*. The Chr14:109.994–114.751 Mb QTL for locomotion contains a stop loss, three frameshift variants, and 9 missense variants in *Slitrk6*.

### 3.6. PheWAS Analysis of the Genes within QTLs

Another method to identify candidate genes is to leverage data generated in another population or species. Phenome-wide association studies (PheWAS) take a gene or variant of interest and find all reported associations in GWAS datasets. A number of these GWAS tools exist, using either different methods, or different human cohorts (https://atlas.ctglab.nl/PheWAS, http://pheweb.sph.umich.edu/, accessed on 2 February 2022).

Mouse QTL mapping has high power but low precision (i.e., we can detect a QTL, but do not know which of tens or hundreds of genes is causal), whereas human GWAS has low power but high precision (tens or hundreds of thousands of individuals are needed, but candidate regions are often smaller). By combining the power of mouse QTL mapping and the precision of human PheWAS, we can do more than both individually. Candidate genes might show up in our analysis here that did not show up in our above analysis for several reasons, the most common being that gene expression was not measured in the relevant cell type or timepoint.

The QTL for vertical activity (Chromosome3 51.723–56.473 Mb) includes several genes with relevant psychiatric, neurological, and cognitive PheWAS hits. *Maml3* is associated with alcohol dependence [68] and depression (Figure 7 and Table 5) [69]. *Cog6* has significant associations with depressive symptoms [70] and worrier/anxious feelings [50]. *Nbea* is associated with nervous feelings [50] and alcohol dependence [68].

All the three genes present in the Chr5 4.468–5.172 Mb QTL (handling-induced convulsions, ethanol response) show significant PheWAS hits for psychiatric traits. *Fzd1* (location of cis-eQTL: Chromosome5 4.753 Mb) is significantly associated with major depressive disorder [71]. In the QTL residing in Chromosome5, the peak 99.801–101.331 Mb region contains the genes *Hnrnpd* and *Lin54,* which show the highest number of relevant pheWAS hits. *Lin54* is associated with conditions such as loneliness, anxiety, tension, and sleep related phenotypes (Figure 8 and Table 6) [50,72,73].

*Cadm1* (Location of *cis*-eQTL: Chromosome9 47.550 Mb) gene was found significantly associated with schizophrenia and other psychiatric disorders [71,74]. Among the genes with *cis*-eQTLs in Chromosome11, *Elac2* (location of *cis*-eQTL: Chr11 64.988 Mb) and *Arghap44* have most significant phenotype associations with schizophrenia/bipolar disorder [74,75]. The QTL for locomotion in the centre (Chr14 109.994–114.751 Mb) shows one gene with a PheWAS hit. *Slitrk6* is significantly associated with Parkinson’s disease [76] and bipolar disorder [75], as well as has significant associations with various psychiatric traits including anxiety [50], nervous feelings [50] and alcohol dependence [68] (Appendix A).

## 4. Discussion

Here, we have demonstrated that old data in populations of isogenic strains can be reanalysed to identify novel genetic associations containing novel candidate genes. Of particular interest is *Slitrk6* on Chr14. *Slitrk6* (SLIT and NTRK Like Family Member 6) is a protein coding gene. Our analysis strongly shows that abnormality in *Slitrk6* is implicated in disrupted locomotor behaviour. The presence of cis-eQTL implies that a variant in this gene is affecting its expression and the gene is under its own regulation. Being part of a network in the striatum, which is significantly involved in abnormal locomotory behaviour and abnormal voluntary movement, increases the plausibility. This gene has evidence of both altered gene expression and protein structure/function, and human PheWAS analysis shows that this gene is involved in various neuropsychiatric and neurological phenotypes. The *Slitrk* family have been previously mentioned as prominent candidate genes involved in neuropsychiatric disorders [77]. The members of the *Slitrk* family have been shown to be widely expressed in the central nervous system, with partially overlapping yet differential patterns of expression [78]. It is worth noting that this gene along with the other candidates have not been reported in the original study.

Another prominent finding is *Cadm1* (Cell adhesion molecule 1), which is a member of the immunoglobulin superfamily and present on Chromosome9. Our analysis shows the presence of a cis-eQTL for this gene, and variants in the human gene are associated with schizophrenia. *Cadm1* knockout mice show anxiety-like behaviour in the open-field and light-dark transition tests, as well as motor coordination and gait impairments in rotarod and footprint tests [63]. The role of CADM1 in relation to prefrontal brain activities, inhibition function, and ADHD, indicating a potential “gene–brain–behaviour” relationship was shown previously by research that evaluated the association of CADM1 genotype with ADHD, executive function, and regional brain functions [79]. Studies show a connection between ADHD and pain tolerance [80], and adults with ADHD are comparatively more sensitive to pain. In such cases, dopamine agonists such as methylphenidate (MP) may exert antinociceptive properties [81] and normalize pain perception. Adults and children with ADHD exhibit motor regulation problems which are in turn associated with pain levels [82].

We discovered a novel QTL that regulates and handles induced convulsions after ethanol treatment (BXD_11635) on Chr5:4.468–5.172. Only three genes are within the confidence interval for this QTL, two of which, *Fzd1* and *Cdk14*, have cis-eQTL and predicted deleterious variants. Interestingly, *Cdk14* is regulated by the anticonvulsant drug valproic acid [83,84,85] and is up-regulated in malaria patients who experience febrile convulsions [41,51,86].

## 5. Conclusions

In this analysis, using GeneNetwork.org (http://www.genenetwork.org/, accessed on 2 February 2022), we have demonstrated the plausibility of using new tools to re-examine older data to investigate candidate genes relevant to addiction research. We used families of isogenic strains of mice to not only go back and discover new drug-related phenotype–genotype associations that were not previously found, but also find highly plausible candidate genes within these novel QTL. Of these genes, many were found to have implications for phenotypes of interest in addiction research, as well as translatability across mouse and human datasets. This sort of investigation is key in the study of addiction-related illnesses, since these diseases are complex and polygenic in nature, and also possess explicit environmental components.

## Figures and Tables

**Figure 1 genes-13-00614-f001:**
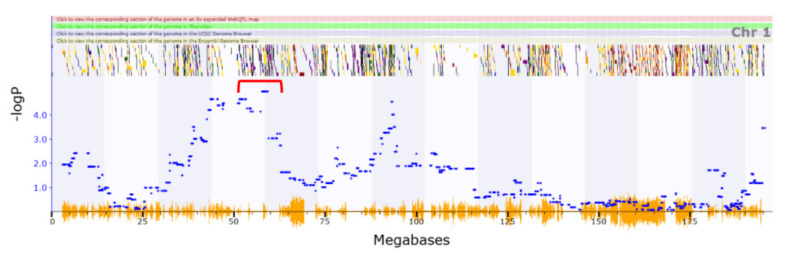
Phenotypic traits associated with locomotion map to a QTL region on chromosome 1. The peak in this region occurs between Chr1:37.67–78.94 as indicated by the red bracketed bar above. The multi-coloured lines corresponding to the dots are genes contained in this region. Blue dots correspond to areas with specific −logP scores relating to the phenotype.

**Figure 2 genes-13-00614-f002:**
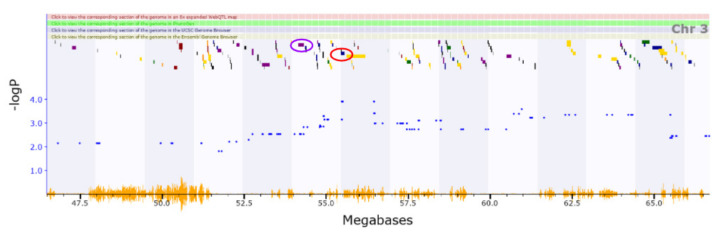
(55.52 Mb) shown in the red circle, and Trpc4 shown in the purple circle (54.27 Mb), which both have cis-eQTLs in this region. Although these genes have not been annotated for this specific phenotype, they are implicated in previous studies with associated behaviors.

**Figure 3 genes-13-00614-f003:**
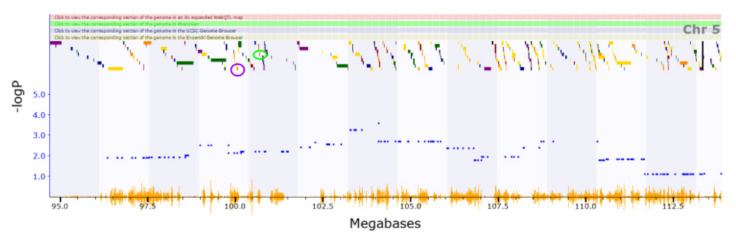
A QTL corresponding with locomotion in response to cocaine has ten genes with cis-eQTLs for Trait ID BXD_11487. A QTL containing Enoph1 shown in the purple circle (100.062 Mb) has been implicated in previous research for its role in stress reactivity. Coq2 shown in the green circle (100.654 Mb) has been investigated for its contribution to neurodegenerative disorders. Although these two genes are great candidates for genes of interest, these aren’t the only applicable. Other relevant genes in this section have not been reported yet by other studies.

**Figure 4 genes-13-00614-f004:**
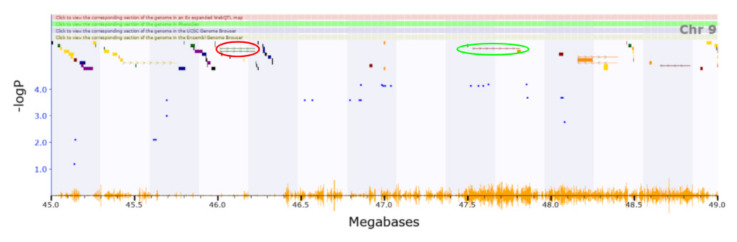
Shown in the red circle (46.22 Mb) is involved in regulating sleep behaviors. Cadm1 shown in the green circle (47.55 Mb) is associated with increased anxiety, impaired social and emotional behaviors, and disrupted motor coordination in knockout mice.

**Figure 5 genes-13-00614-f005:**
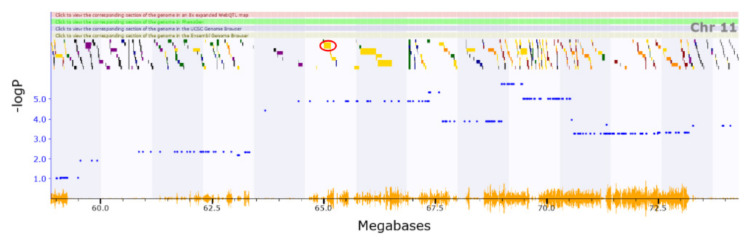
A QTL for motor activity in response to cocaine revealed five genes with cis-eQTLs. Arhgap44 shown in the red circle (65.01 Mb) associates with abnormal motor learning, abnormal response to novel objects, increased grooming behavior, and hypoactivity.

**Figure 6 genes-13-00614-f006:**
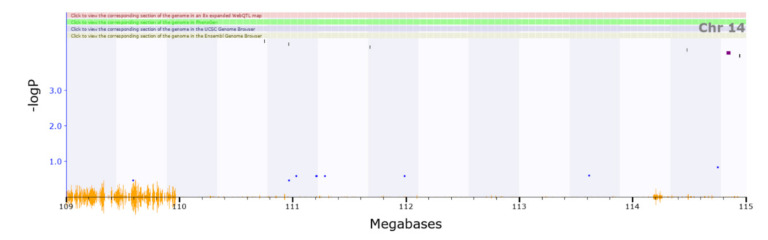
There is a single gene with a cis-eQTL corresponding to the QTL region on chromosome 14. Slitrk6 (109.23 Mb), loss of this gene in mice has been associated with impaired locomotory behavior as well as altered responses to novel environmental cues.

**Figure 7 genes-13-00614-f007:**
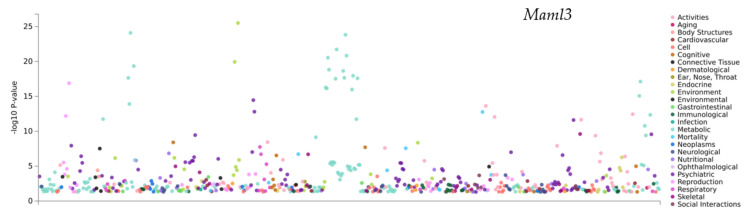
QTLs linked with murine phenotypes gain precision with the use of relevant PheWAS hits in a GWAS atlas. An example for this is for vertical activity (Chr 3 51.723-56.473 Mb) includes a number of genes with relevant psychiatric, neurological and cognitive PheWAS hits. Maml3 is associated with alcohol dependence and depression.

**Figure 8 genes-13-00614-f008:**
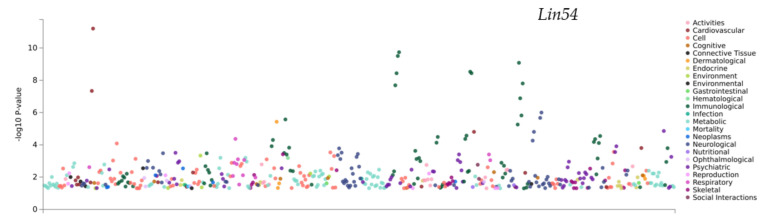
Combining PheWAS hits in GWAS atlases with BXD data allow for more robust screening of variants that affect phenotypes. The QTL at Chr 5 peaks at 99.801–101.331 Mb and contains the genes Hnrnpd and Lin54, which show the highest number of relevant pheWAS hits. Lin54 is listed by trait above and has been previously associated with psychiatric phenotypes.

**Table 1 genes-13-00614-t001:** Summary of novel QTL, not found at the significant or suggestive level in the original paper by Philip et al. [39]. The position of the QTL, a summary of the phenotypes within that QTL, and relevant phenotypes found in other studies are shown. Details of all identified QTL are in Appendix A.

Chromosome	QTL Confidence Interval (Mb)	Summary of Phenotype	Relevant Behaviour Phenotype	PMID of Relevant Phenotype
Chr1	37.671–78.94	Locomotion	Loss of righting induced by ethanol	8974320
Chr1	37.671–78.94	Locomotion	Vertical clinging	10086232
Chr1	68.798–80.329	Cocaine and locomotion	Loss of righting induced by ethanol	16803863
Chr1	91.214–99.884	Vertical activity	Loss of righting induced by ethanol	16803863
Chr3	51.723–56.473	Vertical activity		
Chr7	97.466–104.149			
Chr12	82.859–96.105	BXD_11407		
Chr14	109.994–114.751	BXD_12023		
Chr15	71.035–77.148	Motor coordination, anxiety	Abnormal fear/anxiety-related behaviour	10556431

**Table 2 genes-13-00614-t002:** QTL located at Chr3:53.667–54.942Mb for vertical activity contains predicted deleterious variants in 10 genes.

Number of Variants	Type of Variant	Gene
**2**	Missense	*Ccdc169*
**1**	Missense	*Ccna1*
**1**	In-frame insertion	*Dclk1*
**2**	Frameshift	*Frem2*
**1**	Stop loss	*Frem2*
**9**	Missense	*Frem2*
**1**	Frameshift	*Mab21l1*
**6**	Missense	*Mab21l1*
**9**	Frameshift	*Nbea*
**2**	In-frame deletions	*Nbea*
**19**	Missense	*Nbea*
**3**	Stop loss	*Nbea*
**4**	In-frame deletions	*Postn*
**1**	In-frame insertions	*Postn*
**6**	Missense	*Postn*
**1**	Start loss	*Postn*
**1**	Frameshift	*Spg20*
**8**	Missense	*Spg20*
**1**	Stop gain	*Spg20*
**1**	Stop loss	*Spg20*
**1**	Missense	*Trpc4*

**Table 3 genes-13-00614-t003:** The Chr5:100.164-100.895Mb QTL for cocaine related phenotypes contains predicted deleterious variants in 8 genes.

Number of Variants	Type of Variant	Gene
**3**	Frameshift	*Cops4*
**3**	Missense	*Cops4*
**1**	Missense	*Enoph1*
**1**	Stop gain	*Enoph1*
**1**	Frameshift	*Hnrnpd*
**5**	Missense	*Hnrnpd*
**3**	Missense	*Hnrnpd*
**1**	Splice donor	*Hnrnpd*
**2**	Frameshift	*Hpse*
**1**	In-frame insertion	*Hpse*
**13**	Missense	*Hpse*
**1**	Splice donor	*Hpse*
**3**	Missense	*Lin54*
**1**	Frameshift	*Sec31a*
**1**	In-frame deletion	*Sec31a*
**1**	Splice donor	*Sec31a*
**1**	Missense	*Tmem150c*
**1**	Splice donor	*Tmem150c*

**Table 4 genes-13-00614-t004:** The QTL located in chromosome 9 at 45.671-48.081Mb for mechanical nociception contains predicted deleterious variants in 3 genes.

Number of Variants	Type of Variant	Gene
**1**	Frameshift	*4931429L15Ri*
**2**	Missense	*4931429L15Ri*
**1**	Stop gain	*4931429L15Ri*
**2**	Frameshift	*Cadm1*
**5**	Missense	*Cadm1*
**1**	In-frame deletion	*Cep164*
**3**	Missense	*Cep164*
**1**	Stop loss	*Cep164*

**Table 5 genes-13-00614-t005:** QTLs linked with murine phenotypes gain precision with the use of relevant PheWAS hits in a GWAS atlas. An example for this is for vertical activity (Chr 3 51.723-56.473 Mb) includes a number of genes with relevant psychiatric, neurological and cognitive PheWAS hits. Maml3 is associated with alcohol dependence and depression.

atlas ID	PMID	Year	Domain	Trait	*p*-Value	N
4314	30643251	2019	Psychiatric	Ever smoked regulary	3.47 × 10^−15^	262990
3654	31427789	2019	Psychiatric	Smoking status: Never	2.45 × 10^−12^	384964
4327	30643256	2019	Psychiatric	Well-being spectrum	2.73 × 10^−10^	2311184
4322	30643256	2019	Psychiatric	Depressive symptoms (univariate)	3.58 × 10^−10^	1067913
4313	30643251	2019	Psychiatric	Age of initiation of regular smoking	1.16 × 10^−8^	632802
3425	31427789	2019	Psychiatric	Ever smoked	9.55 × 10^−8^	385013
3236	31427789	2019	Psychiatric	Past tobacco smoking	1.02 × 10^−7^	355594
4274	30846698	2019	Psychiatric	Short sleep	2.64 × 10^−7^	411934
3261	31427789	2019	Psychiatric	Alcohol intake frequency	3.80 × 10^−7^	386082
4326	30643256	2019	Psychiatric	Depressive symptoms (MA GWAMA)	5.12 × 10^−7^	1067913
56	27089181	2016	Psychiatric	Depressive symptoms	1.72 × 10^−6^	161460
3796	29942085	2018	Psychiatric	Depressive symptoms	1.75 × 10^−6^	381455
3235	31427789	2019	Psychiatric	Current tobacco smoking	2.71 × 10^−6^	386150
4293	30718901	2019	Psychiatric	Depression	3.19 × 10^−6^	500199
3268	31427789	2019	Psychiatric	Alcohol intake versus 10 years previously	3.41 × 10^−6^	357907
4171	29970889	2018	Psychiatric	Loneliness	3.44 × 10^−6^	445024
4170	29970889	2018	Psychiatric	Loneliness (MTAG)	3.72 × 10^−6^	487647

**Table 6 genes-13-00614-t006:** Combining PheWAS hits in GWAS atlases with BXD data allow for more robust screening of variants that affect phenotypes. The QTL at Chr 5 peaks at 99.801–101.331 Mb and contains the genes Hnrnpd and Lin54, which show the highest number of relevant pheWAS hits. Lin54 is listed by trait above and has been previously associated with psychiatric phenotypes.

atlas ID	PMID	Year	Domain	Trait	*p*-Value	N
4327	30643256	2019	Psychiatric	Well-being spectrum	1.40 × 10^−5^	2311184
3998	29500382	2018	Psychiatric	Tense	1.23 × 10^−5^	263635
3291	31427789	2019	Psychiatric	Tense	2.80 × 10^−4^	374129
4293	30718901	2019	Psychiatric	Depression	3.12 × 10^−4^	500199
4325	30643256	2019	Psychiatric	Neuroticism (MA GWAMA)	3.94 × 10^−4^	523783
3798	29942085	2018	Psychiatric	Worry subcluster	5.57 × 10^−4^	348219
4087	29255261	2018	Psychiatric	Neuroticism	8.90 × 10^−4^	329821
4322	30643256	2019	Psychiatric	Depressive symptoms (univariate)	1.05 × 10^−3^	1067913
3301	31427789	2019	Psychiatric	Seen doctor (GP) for nerves, anxiety, tension or depression	1.11 × 10^−3^	383771
4321	30643256	2019	Psychiatric	Neuroticism (univariate)	1.12 × 10^−3^	523783
4326	30643256	2019	Psychiatric	Depressive symptoms (MA GWAMA)	1.27 × 10^−3^	1067913
3302	31427789	2019	Psychiatric	Seen a psychiatrist for nerves, anxiety, tension or depression	2.48 × 10^−3^	384700
3745	31427789	2019	Psychiatric	Happiness and subjective well-being—General happiness	2.83 × 10^−3^	126132
4011	29662059	2018	Psychiatric	Broad depression	3.34 × 10^−3^	322580
4013	29662059	2018	Psychiatric	Major depressive disorder (ICD-coded)	3.46 × 10^−3^	217584
4269	30867560	2019	Psychiatric	Neuroticism general factor	3.84 × 10^−3^	270059
3230	31427789	2019	Psychiatric	Morning/evening person (chronotype)	4.42 × 10^−3^	345148

## Data Availability

All data supporting this article can be accessed and reanalyzed using GeneNetwork.org.

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
