# Peer review of "New Insights on Gene by Environmental Effects of Drugs of Abuse in Animal Models Using GeneNetwork"

_genes, 2022, doi:10.3390/genes13040614_

Round 1

Reviewer 1 Report

The manuscript "Old data and friends improve with age: new insights with the updated tools of GeneNetwork" by Chunduri and colleagues describes the utilization of older datasets via their GeneNetwork portal. Despite that the study direction and the general phylosophy of the manuscript can be agreed upon (applying new methods to old data to extract new insights), the tool itself does not constitute a sufficient enough improvement over their previous publication on JOSS (https://joss.theoj.org/papers/10.21105/joss.00025). Below, some more specific points:

Major:

The genenetwork.org phylosophy described in the introduction is certainly aiming at the right direction, namely the concept of "deeper data mining" using already existing datasets and new methods. However, the web portal shows a counterintuitive organization of available data, with limited explanation to the explorable datasets. One suggestion for the authors would be to further subdivide the "molecular trait" data type. First of all, also the genotype (which corrently is secluded in a separate menu entry) is a molecular trait. Secondly, I would distinguish and separate, for higher organization and accessiblity, quantitative data into: transcritpomics (mRNA, microRNA, lncRNA), metabolomics, proteomics and lipidomics. This would project the website and the project in the next multiomics era by clarifying the multiple data types contained in the dataset.

The design and the possibilities provided by the geneNetwork web portal are limited, as it makes impossible to check the same feature (e.g. a gene) across different datasets. For example, a search of the gene "MYCN" provides simply a table of the Affymetrix probeset ids associated to it. Alterantively, it is possible to search the gene MYCN via the "select and search" interface, but this will allow to retrieve only the probeset ids within a specific dataset (e.g. Hippocampus mRNA) and then certify, via e.g. the "Correlations" analysis interface, a correlation between the three probesets of the same gene in the selected dataset. It is therefore impossible to, for example, detect potential coexpressors of the MYCN gene across different datasets, or perform any useful gene-centric analysis. Many successful tools can inspire a potential design upgrade for GeneNetwork, for example the R2 tool by Jan Koster at https://hgserver1.amc.nl/cgi-bin/r2/main.cgi or the MSKCC cBio portal at https://www.cbioportal.org/.

The enormous potential of combining datasets, old and new, should be adopted by the GeneNetwork portal. One example is the inclusion of the vast collections of publicly available reference datasets for human healthy tissues contained in the GTEX database. This includes transcriptomics and genomics data, and also provides tons of information regarding the association between datasets (e.g. cis- and trans-eQTL analyses). Several other datasets, including QTL data which GeneNetwork seems to be orbiting around, are available and should be included by an improved GeneNetwork portal (see e.g. the GWAS catalogue at https://www.ebi.ac.uk/gwas/)

In general, the authors provide several figures (1 to 8) ant tables (1 to 4) to describe different QTL analyses, with one or several genes. In fact, the entire figure and table collection shows different examples of QTL analysis, making the multiomics potential of GeneNetwork, and the notion of "network" itself, fade under the exclusivity of one analysis type. This seems simply adding more examples of the same analysis framework that was described in the original GeneNetwork paper (https://joss.theoj.org/papers/10.21105/joss.00025) and does not provide any of the features foreshadowed in the introduction or required by an actual incremental update, namely the utilization of multiple genomics datasets with novel methods.

Minor:

The abstract does not contain any explanation on the term "GeneNetwork" which is used in the title.

The term "genometype" is still (unfortunately) not very common in the community of geneticists (despite it having been coined almost 10 years ago https://www.science.org/doi/epdf/10.1126/scitranslmed.3003380), so I suggest quickly defining it in the intro, in order to let a wider audience grasp its connotations (and not mistake it for "genotype").

Line 31: do the authors mean "irreproducible" rather than "irreducible"?

Author Response

Reviewer 1:

Thank you for your useful comments! Our manuscript is a new analysis looking at gene by environmental effects but isn’t intended as a manuscript just for GeneNetwork.org. We are not developers as we are more focused on the biological questions regarding drug abuse and behavior, but we passed these comments along to the GeneNetwork team to aid in its development.

Addressing some of your comments, our responses are in italics:

The design and the possibilities provided by the GeneNetwork web portal are limited, as it makes impossible to check the same feature (e.g. a gene) across different datasets.

The global search tool at the top of the website can search for a gene or phenotype (dependent on the drop down window to the left) across all organisms and data sets.

One example is the inclusion of the vast collections of publicly available reference datasets for human healthy tissues contained in the GTEX database.

There are reference datasets in GN that include the GTEX information. There are three versions: v3, v5, and v8.

The abstract does not contain any explanation on the term "GeneNetwork" which is used in the title.

The term GeneNetwork has been changed to GeneNetwork.org throughout the manuscript. The title has also been changed to “New insights on gene by environmental effects of drugs of abuse in animal models using GeneNetwork.org.”

The term "genometype" is still (unfortunately) not very common in the community of geneticists (despite it having been coined almost 10 years ago https://www.science.org/doi/epdf/10.1126/scitranslmed.3003380), so I suggest quickly defining it in the intro, in order to let a wider audience grasp its connotations (and not mistake it for "genotype").

Genometype has been described in the introduction as “The genometype refers to all genotype states across the organism. Different strains in the BXD family might share the same genotype at a specific location, but the different strains are different genometypes.”

Line 31: do the authors mean "irreproducible" rather than "irreducible"?

“Irreducible has been changed to “irreproducible” in line 35 (what was line 31). “There is still the problem of irreproducible datasets: for example, if a sample from a particular outbred cohort is found to be an outlier during data analysis, there is no way to go back to that genometype and remeasure the phenotype.”

Reviewer 2 Report

The authors presented novel application of QTL analysis using GeneNetwork.org tool. The material well prepared. It is easy to read though the references and terminology should be updated. I believe the paper will be interesting to the readers.

However I have some remarks demanding revision.

Major:

First, I suggest change title to reflect the paper content. Current title is quite artistic, but it is worthy add wording like ‘animal models’, ‘behavioural phenotypes’ or like that.

Moreover, the word GeneNetwork is occupied already for gene network analysis tools.

See

http://wwwmgs.bionet.nsc.ru/mgs/gnw/

and publications there (GeneNet database)

see also

https://genenetwork.nl/

It is important issue related to copyrights.

I suggest use name  ‘GeneNetwork.org’

http://genenetwork.org/

Bulk citations - lines 52-54: ‘[4,5,6,7,8,9,10], immunology [11,12,13,14,15],...’ - such citation style is not acceptable. Please cite 1-2, maximum 3 papers together, but not 5-6 as here. Remove redundant references. Or add some phrases commenting about application of these lines to different areas. Otherwise it looks like plagiarism or some commercial advertisements.

Minor.

Need provide all the abbreviations in full. Not all the readers know specific laboratory animal terminology. Comment on BXD

Line 10: ‘mice, such as the BXD’

- add a phrase about BXD family (recombinant inbred strains from a cross of C57BL/6J and DBA/2J mice).

The same remark is on the main text.

Line 13: ‘behavioural phenotypes from Philip et al. 2010’ - try to rephrase, add journal name at least. The citation is not visible in the Abstract. May comment about ‘behavioural phenotypes’ data

Line 14: ‘models (GEMMA and R/qtl2)’ - may give GEMMA in full, comment on methods

Line 29: ‘FAIR Principles’ - give ‘FAIR’ in full

Line 39: ‘The GeneNetwork.org website’ - this is correct name of the tool. MAy add formal web-link too  - like ‘the GeneNetwork.org (http://genenetwork.org)...’

Line 44 ‘BXD family of murine strains’ - add a phrase about BXD family (similar to the line 10 remark). It is now at line 55, but should be given from the beginning.

Line 67: ‘GeneNetwork’ - need name it GeneNetwork.org or write some phrase ‘we will name it GeneNetwork tool’ to give definition to use in the next text.

Line 105: ‘from this (Personal..’ - not correct wording, either give the name of the person, or change the reference.

Line 110: ‘GEMMA’ - give abbreviation in full in the section title. Or write ‘GEMMA method’

Line 116: ‘R/qtl2’ - comment on the method abbreviation similar as for GEMMA (in sama paragraph)

Line 140: ‘A 1.5 LOD or 1.5 -log(p)...’ - give ‘LOD’ in full. Keep sign ‘-‘ with ‘log’ (see line-141-142).

What is used - minimal or ant of these values if they are not equal? (LOD and -log(p) )

Line 189: ‘(H-K; as used previously)’ - add reference here

Table 1 -  not need extra frame around the table.

Figures 1 and similar ones have too small signs, it is hard to see the screenshots. Try to improve figure quality, increase fonts on axis X. At least may remove top lines (I hardly can read ‘click here to view..’ on pink, green top lines in the figure frame- assume this part may be removed)

Frames around all figures and tables could be removed too. I think it is some technical mistake. Figure legend should not be in the figure frame

Line 428: ‘In this analysis using GeneNetwork...’  - add correct link and name it 'GeneNetwork.org'

Author Response

Reviewer 2:

Thank you for the comments on our manuscript. Our responses to your comments are in italics.

First, I suggest change title to reflect the paper content. Current title is quite artistic, but it is worthy add wording like ‘animal models’, ‘behavioural phenotypes’ or like that. Moreover, the word GeneNetwork is occupied already for gene network analysis tools. I suggest use name ‘GeneNetwork.org’ http://genenetwork.org/

We have changed the title of the paper to “New insights on gene by environmental effects of drugs of abuse in animal models using GeneNetwork.org.” The use of the word GeneNetwork has been changed to GeneNetwork.org throughout the paper.

Bulk citations - lines 52-54: ‘[4,5,6,7,8,9,10], immunology [11,12,13,14,15],...’ - such citation style is not acceptable. Please cite 1-2, maximum 3 papers together, but not 5-6 as here.

We have changed the bulk citations to just contain the 3 most relevant papers.

Need provide all the abbreviations in full. Not all the readers know specific laboratory animal terminology. Comment on BXD Line 10: ‘mice, such as the BXD’

- add a phrase about BXD family (recombinant inbred strains from a cross of C57BL/6J and DBA/2J mice). The same remark is on the main text.

BXD has been defined in the abstract and in the main text to “BXD strains are recombinant inbred mouse strains derived from crossing two inbred strains—C57BL/6J and DBA/2J mice.”

Line 13: ‘behavioural phenotypes from Philip et al. 2010’ - try to rephrase, add journal name at least. The citation is not visible in the Abstract. May comment about ‘behavioural phenotypes’ data.

The citation has been reworded to add the journal name to “Philip et al. Genes, Brain & Behavior 2010.”

Line 14: ‘models (GEMMA and R/qtl2)’ - may give GEMMA in full, comment on methods

GEMMA has been defined in the abstract and the methods to “Genome-wide Efficient Mixed Model Association.”

Line 29: ‘FAIR Principles’ - give ‘FAIR’ in full

FAIR has been defined in the introduction to “findability, accessibility, interoperability, and reusability.”

Line 39: ‘The GeneNetwork.org website’ - this is correct name of the tool. MAy add formal web-link too - like ‘the GeneNetwork.org (http://genenetwork.org)...’

The web link was added “The GeneNetwork.org (http://www.genenetwork.org/) website…”

Line 44 ‘BXD family of murine strains’ - add a phrase about BXD family (similar to the line 10 remark). It is now at line 55, but should be given from the beginning.

BXD has been defined in the abstract and in the main text to “BXD strains are recombinant inbred mouse strains derived from crossing two inbred strains—C57BL/6J and DBA/2J mice.”

Line 67: ‘GeneNetwork’ - need name it GeneNetwork.org or write some phrase ‘we will name

it GeneNetwork tool’ to give definition to use in the next text.

The term GeneNetwork has been changed to GeneNetwork.org throughout the manuscript.

Line 105: ‘from this (Personal..’ - not correct wording, either give the name of the person, or change the reference.

Reference has been changed to “this (European Nucleotide Archive project PRJEB45429).”

Line 110: ‘GEMMA’ - give abbreviation in full in the section title. Or write ‘GEMMA method’

GEMMA has been defined in the abstract and the methods to “Genome-wide Efficient Mixed Model Association.”

Line 116: ‘R/qtl2’ - comment on the method abbreviation similar as for GEMMA (in sama

paragraph)

R/qtl2 is characterized by “qtl2 analysis using R software” in the method section.

Line 140: ‘A 1.5 LOD or 1.5 -log(p)...’ - give ‘LOD’ in full. Keep sign ‘-‘ with ‘log’ (see line-141-

142).

LOD was defined as “logarithm of the odds” in line 147.

What is used - minimal or ant of these values if they are not equal? (LOD and -log(p))

The manuscript states that these methods are similar and considered equal.

Line 189: ‘(H-K; as used previously)’ - add reference here

Reference added to text.

Table 1 - not need extra frame around the table. Figures 1 and similar ones have too small signs, it is hard to see the screenshots. Try to improve figure quality, increase fonts on axis X. At least may remove top lines (I hardly can read ‘click here to view..’ on pink, green top lines in the figure frame- assume this part may be removed) Frames around all figures and tables could be removed too. I think it is some technical mistake. Figure legend should not be in the figure frame

Frames around the figures and tables were removed. Unfortunately, the graphs on GeneNetwork.org can not be manipulated. We have made the figures larger as a whole.

Line 428: ‘In this analysis using GeneNetwork...’ - add correct link and name it 'GeneNetwork.org'

GeneNetwork was changed to GeneNetwork.org and the link was added, “In this analysis using GeneNetwork.org (http://www.genenetwork.org/)..”

Reviewer 3 Report

This work, “Old data and friends improve with age: new insights with the updated tools of GeneNetwork” (genes-1602576) presented by David G. Ashbrook et al reports their interesting study regarding reanalysis of old data in populations of isogenic strains. They revealed several novel genetic associations containing novel candidate genes. This is an interesting study worth publication. However, several concerns should be addressed before acceptance.

MAJOR POINTS:

  1. Abstract. “Candidate genes included Slitrk in a Chr 14 QTL for locomotion, found to be part of a coexpression network involved in voluntary movement and associationed with neuropsychiatric phenotypes; and Cdk14, one of only 3 genes in a Chr 5 QTL for handling induced convulsions after ethanol treatment, that is regulated by the anticonvulsant drug valproic acid.”---confusing sentence.
  2. Language. This paper should be polished by a professional English editing company.

MINOR POINTS

  1. line 289. “other studies”---refs should be added.

2. line 332. Table2 (also see 343 Table 3…and Table 4)---the IDs of genes should be added. 

Author Response

Reviewer 3:

Thank you for the comments on our manuscript. Our responses to your comments are in italics.

Abstract. “Candidate genes included Slitrk in a Chr 14 QTL for locomotion, found to be part of a coexpression network involved in voluntary movement and associationed with neuropsychiatric phenotypes; and Cdk14, one of only 3 genes in a Chr 5 QTL for handling induced convulsions after ethanol treatment, that is regulated by the anticonvulsant drug valproic acid.” ---confusing sentence.

The abstract has been changed to clarify the sentence. The sentence has been changed to multiple sentences to include, “Candidate genes included Slitrk6 and Cdk14. Slitrk6, in a Chromosome14 QTL for locomotion, was found to be part of a coexpression network involved in voluntary movement and associated with neuropsychiatric phenotypes. Cdk14, one of only 3 genes in a Chromosome5 QTL, is associated with handling induced convulsions after ethanol treatment, that is regulated by the anticonvulsant drug valproic acid.”

Language. This paper should be polished by a professional English editing company.

The manuscript has been reviewed for grammar and spelling, and suggested edits have been made.

Line 289. “other studies”---refs should be added.

We could not find the passage mentioned in your comment. Could you please provide more information so that we can address it?

Line 332. Table2 (also see 343 Table 3…and Table 4)---the IDs of genes should be added.

Which IDs should be used for the genes? Could you please provide more information so that we can address this?

Round 2

Reviewer 1 Report

I still think the authors did not write a sufficient enough advancement over previous work to justify a publication, but the paper has improved with revision and the authors have addressed all the raised points

Reviewer 3 Report

After this round of revision, the quality of the manuscript has been greatly improved, making it ready for acceptance.